# A Feasible and Promising Strategy for Improving the Solar Selectivity and Thermal Stability of Cermet-Based Photothermal Conversion Coatings

**DOI:** 10.3390/ma15196847

**Published:** 2022-10-02

**Authors:** Xiaobo Wang, Yabin Kang, Xiaopeng Yuan, Dianqing Gong, Kewei Li

**Affiliations:** 1Department of Physics and Electronic Engineering, Jinzhong University, Jinzhong 030619, China; 2College of Materials Science and Engineering, Taiyuan University of Technology, Taiyuan 030024, China

**Keywords:** solar selective absorbing coating, thermal stability, selectivity

## Abstract

A novel multilayer, solar selective absorbing coating that contains lamellar-distributed nanoparticles in its cermet-absorbing sublayers has been fabricated using ion-source-assisted cathodic arc plating. The multilayer coating shows an outstanding selectivity, i.e., a high solar absorptance (0.909), yet it has a low thermal emittance (0.163). More importantly, the long-term thermal stability tests demonstrate that the lamellar-structured absorbers can remain stable, even when annealed at 500 °C for 1000 h in ambient air. The coating’s enhanced selectivity and thermal stability were attributed to the formation of lamellar-distributed nanoparticles in the absorbing sublayer, which form many asymmetric Fabry–Pérot cavities. In this case, the light would be held in the Fabry–Pérot cavities and thus boost the absorptivity due to the increase in interaction time. Meanwhile, the unique distribution of the nanoparticles is also beneficial for enhancing the surface plasmon resonance absorption, and thus promoting the increase in solar selectivity. Furthermore, the excellent thermal stability is ascribed to the existence of amorphous matrices, which separate and seal the nanoparticles into honeycomb shells. In this case, the atomic diffusion in the nanoparticles would be significantly retarded as the amorphous matrices can remain stable below the crystallization temperatures, which can effectively slow down the growth and agglomeration of the nanoparticles.

## 1. Introduction

With the skyrocketing consumption of traditional fossil fuels, such as coal, oil, and natural gas, we are confronted with an energy crisis as these energy resources are non-renewable and are quickly diminishing. Meanwhile, the enormous use of fossil fuels has deteriorated the environment by polluting the air and water, creating toxic wastes, and causing global warming. One remedy for this problem is the implementation of a renewable energy policy, which has been considered as the most effective way of solving the energy crisis. Among the existing renewable energies, the thermal use of solar energy has attracted increasing attention and numerous investigations due to its advantages of unlimited availability, being clean and pollution-free, and being renewable [1]. To boost its photothermal conversion efficiency, the most effective way to utilize solar energy is to use solar selective absorbing coatings that are capable of converting sunlight into heat efficiently due to their high solar absorptance and low thermal emittance [2,3,4]. The performance of a solar selective absorbing coating directly determines the efficiency of a photothermal system. Thus, developing solar selective absorbers with simultaneous high selectivity and excellent thermal stability is crucial to the commercialization of solar thermal utilization.

Cermet coatings that consist of extremely fine metal particles embedded in dielectric matrices are the most widely applied solar selective absorbers due to their tunable microstructures and spectral characteristics [5,6]. However, the fine metals particles are naturally prone to coarsening when working at elevated temperatures, which would inevitably induce deterioration in an optical performance [7,8,9]. Thus, the performance of cermet solar selective absorbing coatings is determined by the thermal tolerances of their nanoparticles.

Hitherto, several strategies including alloying and replacing the metal nanoparticles with alternatives (i.e., nitrides, oxides, and high-entropy alloys) have been attempted to enhance the thermal stability of the nanosized metal particles. Wang et al. developed WTi-Al_2_O_3_ cermet-based absorbing coatings that doped W nanoparticles with Ti elements, resulted in an absorptance of ~93% and a low emissivity of 10.3%, even after annealing at 600 °C for 840 h in a vacuum [7]. Meanwhile, Liu et al. designed a CrAlO-based solar selective absorbing coating in which the metal nanoparticles were replaced by Cr_2_Al intermetallics. The multilayer coating showed a high solar absorptance of 0.924 and a thermal emittance of 0.21, as well as excellent thermal stability, with a selectivity of 0.919/0.225 even when annealed at 700 °C for 2 h in ambient air [10]. Li et al. reported a solar selective absorber that substituted metal particles with TiN, and it not only showed a high solar absorption of ~95% and low thermal emission of ~3%, but it also exhibited excellent stability over a temperature range of 100–727 °C for 5 h in a vacuum [11]. Further, our previous investigations found that using transition metal nitrides as the absorbing units can result in simultaneously high selectivity and outstanding thermal stability [12]. To a certain extent, the thermal tolerance of nanoparticles can be improved by enhancing thermodynamic stability. Nevertheless, current enhancements cannot meet the requirements for high-temperature atmosphere applications [7].

It is worth noting that nanoparticles are randomly distributed in amorphous structured cermet matrices, which facilitates agglomeration at high temperatures and thus accelerates the deterioration of the cermet sublayers. However, agglomeration-induced degeneration is caused by the random distribution of nanoparticles rather than thermodynamic stability, i.e., the reported cermet-based solar selective absorbing coatings have not taken full advantage of the nanoparticles. In this work, we propose a novel strategy that can simultaneously stabilize the nanoparticles and boost selectivity by introducing lamellar-distributed nanoparticles into the absorbing sublayers using ion-source-assisted cathodic arc plating. The optical properties and microstructure of the multilayer coating have been investigated using spectrophotometers, TEM, and GI-XRD. Special attention has been paid to the selective absorbing mechanism of this unique structure, as well as to its thermal stability at high temperatures in ambient air.

## 2. Materials and Methods

### 2.1. Sample Preparation and Annealing

A series of Cr/AlCrN/AlCrON/AlCrO cermet-based solar selective absorbing coatings was deposited on metallographically polished stainless-steel substrates (model 304, dimensions of 30 mm × 40 mm) using a cathodic arc ion plating system. High purity Al_70_Cr_30_ (99.99%) and Cr (99.99%) targets were used as the source materials. The Ar/N_2_/O_2_ gas mixtures were employed to prepare the AlCrN, AlCrON, and AlCrO sublayers using the AlCr target. The pressure of the cathodic arc chamber was maintained at 3 × 10^−1^ Pa during deposition. The detailed deposition parameters are described in the previous work [13].

The thermal stability of the multilayer coating was evaluated in ambient air in a muffle furnace. For the annealing procedure, 5 pieces of the coating were set in the center of the muffle furnace and then heated to 500 °C with a heating rate of 10 °C/min. The temperature accuracy of the muffle furnace was ±5 °C. The thermal stability of the multilayer coating was mainly determined by the coarsening behavior of the nanoparticles and the stability of the structure in the absorbing sublayer. In the present study, the aim was to test the thermal stability of the AlCrON and AlCrN sublayers. Then, the samples were cooled down to room temperature naturally for the optical measurements after an interval of 100 h (denoted as one cycle in the present study), followed by repeating the annealing cycle.

### 2.2. Characterization

The XRD patterns were acquired by a high-resolution Philips X′Pert MRD X-ray diffractometer using Cu Kα radiation (λ = 0.154056 nm) with a grazing incidence angle of 0.3–1.5°. Scanning electron microscopy (SEM, JSM-IT300, JEOL Ltd., Tokyo, Japan) was used to explore the surface and cross-sectional topography of the multilayer coatings. The cross-sectional TEM images of the multilayer coatings were acquired using transmission electron microscopy (TEM, Tecnai G^2^ F20 S-TWIN, Hillsboro, OR, USA) to identify the microstructural features of the multilayer coatings. Selected area electron diffraction (SAED) and EDS mapping analysis were also performed to investigate the evolution of the element distribution during the long-term annealing. To prepare the cross-sectional TEM foils, a focused ion beam (FIB) milling method was employed using a PIPS-Model 691 and a Tescan Lyra 3 XMU FEG/SEM-FIB microscope (Tescan, Brno, Czech Republic). High-angle annular dark-field scanning transmission electron microscopy (HAADF-STEM) imaging analysis was carried out using a Talos F200X microscope (Thermo Fisher Scientific, Waltham, MA, USA) at an accelerating voltage of 200 kV.

### 2.3. Optical Properties Tests

The reflectanceI) solar spectrum range (0.3–2.5 μm) of the coating was obtained by a Shimadzu UV3600 UV/VIS/NIR spectrophotometer with an integration sphere (module 150 mm), while the reflectance spectra in the range of 2.5–25 μm were measured by a Bruker Tensor 27 Fourier Transform spectrometer equipped with an integrating sphere (A562-G/Q) using a gold plate as a standard. Then, the absorptance α and emittance ε were obtained by Equations (1) and (2):(1)$$\alpha =\frac{\int_{0.3}^{2.5}I_{s}\left(\lambda \right)\left[1-R\left(\lambda \right)\right]d\lambda }{\int_{0.3}^{2.5}I_{s}\left(\lambda \right)d\lambda }$$
(2)$$\varepsilon =\frac{\int_{2.5}^{25}I_{b}\left(\lambda \right)\left[1-R\left(\lambda \right)\right]d\lambda }{\int_{2.5}^{25}I_{b}\left(\lambda \right)d\lambda }$$
where *λ* is the wavelength, *I_s_*(*λ*) is the solar radiation power at AM1.5, *R*(*λ*) is the spectral reflectance of sample, and *I_b_(λ)* is the spectral black body emissive power at room temperature.

The thermal stability of the solar selective absorbing coating was evaluated by the performance criterion (PC), which was calculated from the obtained α and ε according to the following Equation (3):
PC = −Δα + 0.5ε(3)
where Δα = α_(aged)_ − α_(unaged)_ and Δε = ε(_aged_) − ε(_unaged_). Once the PC increases to be higher than 0.05, the solar selective absorbing coating is identified as having failed.

### 2.4. Numerical Simulations

The reflection spectra of the multilayer coatings in the wavelength range of 0.3–2.5 μm were computed using the commercial FDTD method (Lumerical software). The refractive index values of the AlN and Cr_2_N nanoparticles were obtained from [14]. The z-direction was defined as a perfectly matched layer, while the x- and y-directions were denoted as periodic boundaries. The mesh sizes along all three directions were set as 1 nm during the reflectance calculation. In addition, the broadband fixed angle source technique was employed in the FDTD simulations. The scattering and absorption cross-sections of nanoparticles and nanoparticle arrays were calculated using a total-field scattered field source as the incident light. The boundary conditions along the *x*-, *y*-, and *z*-directions were all defined as perfectly matched layers. The mesh sizes along the three directions were set to be as small as 0.5 nm.

## 3. Results

### 3.1. Optical Properties and Thermal Stability of the Multilayer Coatings

Figure 1 shows the UV-visible-IR reflectance spectra of the solar selective absorbing coatings before and after annealing. The as-deposited multilayer coatings show relatively low sunlight reflections over the wavelengths of 0.3–1.5 µm, which correspond to the central zone of solar radiation. Meanwhile, the reflectance sharply increased to 0.4 at 2.5 µm and approached 0.97 within the mid-infrared wavelengths. These features indicate that the multilayer coatings should have excellent solar selectivity. Indeed, the calculated optical properties listed in Table 1 suggest a high solar absorptance of 0.909 and a low thermal emittance of 0.163.

More importantly, the reflectance spectra remained the same after annealing at 500 °C in ambient air for 1000 h. Interestingly, the absorptance increased to 0.923 during the subsequent annealing at 500 °C in ambient air. Although the emittance first decreased and then increased, the corresponding PC values were all smaller than the critical standard (0.05), as illustrated in Table 1. Such high selectivity and thermal stability render these multilayer coatings as potential candidates for photo-thermal conversion at high temperatures. Overall, the Cr/AlCrN/AlCrON/AlCrO multilayer coatings deposited in the present paper can sustain superior spectra selectivity at elevated temperatures of up to 500 °C in ambient air, which is better than the performance of commercial solar selective absorbing coatings (<500 °C) [15,16].

### 3.2. Microstructural and Compositional Evolution of the Solar Selective Absorbing Coatings during Long-Term Annealing

In general, the optical properties of the solar selective absorbing coatings are coupled with their phase composition and microstructure characteristics. Therefore, we examined the microstructures of the annealed multilayer coatings. Figure 2 shows the GI-XRD patterns of the multilayer coatings before and after annealing at 500 °C in ambient air for 1000 h. For the as-deposited sample, two distinguishable broadened peaks centered at 42.6° and 44.3° were detected, and they were identified as the (200) and (111) lattice planes of AlN and Cr_2_N, respectively. For the annealed sample, six new discernible peaks were observed as centered at 40.3°, 42.4°, 43.5°, 55.9°, 66.9°, 74.8°, and 81.5°, which are respectively assigned to the (002), (111), (200), (112), (300), (113), and (221) lattice planes of Cr_2_N [7]. It is worth noting that the intensity of the (111) lattice plane of Cr_2_N was significantly enhanced after annealing, likely due to the additional formation of Cr_2_N nanoparticles during the annealing process. In addition, no diffraction peaks corresponding to oxide phases were detected in the as-deposited and annealed coatings. The absence of oxide phases was attributed to the amorphous structure nature of the coatings, as suggested in our previous paper [9].

For further probing of the distribution of AlN and Cr_2_N nanoparticles in the absorbing sublayers, the cross-sectional TEM images of the annealed coatings are presented in Figure 3. It can be discerned from Figure 3a that the annealed coating shows a multilayer structure consisting of a Cr IR reflector, AlCrN/AlCrON absorbers, and an AlCrO anti-reflective sublayer from the substrate to the surface. More importantly, no distinct agglomeration of nanocrystals and structural destruction were observed in the cermet stacked sublayers. Interestingly, a series of alternating sublayers with thicknesses of 10 nm or less was observed in the AlCrON absorber. This type of unique microstructure can be defined as a nanolaminate, which would have a critical effect on solar absorbing behavior and thermal stability. The chemical component mapping via EDS analysis shown in Figure 3b–f demonstrates a spatially inhomogeneous element distribution. This is further confirmed by the EDS line scan shown in Figure 3g. The formation of this unique microstructure, as well as its effect on solar selectivity, will be discussed in the following section.

To gain deep insight into the AlCrON sublayer, cross-sectional STEM and HR-TEM tests were performed on these layered regions, and the detail results are shown in Figure 4. The nanograins in the AlCrON sublayer are identified as AlN and Cr_2_N phases, which is in reasonable agreement with the GI-XRD results. Moreover, the average size of the Cr_2_N is much smaller than that of the AlN since the activation energy it achieves is as high as 296 KJ/mol [9]. It is worth noting that no oxide nanocrystals were detected in the AlCrON absorbing sublayer even though we carried out abundant HR-TEM tests. The absence of oxide nanocrystals may be related to its strong tendency to form an amorphous structure, which merits further study.

The formation of nanolaminates in the multilayer coating is highly related to the characteristic of cathodic arc plating. In general, the reactive N_2_, O_2_, and Ar are distributed uniformly in the chamber during deposition. However, the plasma tends to plume in the normal direction from the cathode surface toward the substrate [17]. The plasma density distribution is more peaked in the axial direction, as shown in Figure 5a, due to the applied axial magnetic field. The heavy chromium ions tend to distribute in region 2, as shown in Figure 5b, due to low diffusion rate, while the aluminum ions are prone to plume regions 1 and 3. In this case, the aluminum ions can completely react with the reactive gas, which would result in the formation of the amorphous matrix in each sublayer. Nevertheless, the chromium ions cannot react completely due to the insufficient amount of gas along the normal direction of the target, and it remained a crystalline structure. When the substrates rotate through these three regions, nanolaminates containing amorphous matrices and nanocrystals would be formed during deposition, as schematically illustrated in Figure 5b. 

## 4. Discussion

As described above, the Cr/AlCrN/AlCrON/AlCrO multilayer coating prepared by cathodic arc ion plating exhibits a high solar selectivity and outstanding thermal stability at 500 °C in ambient air, and it can be considered as a good candidate for high-temperature solar thermal conversion. The enhanced selectivity and thermal stability can be ascribed to the unique distribution of nanoparticles in the AlCrON absorbing sublayer. Thus, the emphasis of this paper is focused on the spectrally selective absorbing mechanism of the layered multilayer coating, as well as the thermal strengthening mechanism.

### 4.1. Spectrally Selective Absorbing Mechanism of the Layered Multilayer Coating

Traditionally, the main absorbing mechanism of cermet coating is related to the localized surface plasmons (LSPs), which are charge density oscillations confined to metallic nanoparticles (sometimes referred to as metal clusters) and metallic nanostructures [18]. The resonance will occur when the wavelength of the incident light is identical to the natural frequency of the nanoparticles. The localized surface plasmon resonance would result in intense light scattering and strong surface plasmon absorption bands [19]. However, the absorptivity of the cermet-based coating was increased during the long-term annealing. Thus, there must be additional absorbing mechanisms contributing to the enhancement in solar absorption.

With the aim of understanding the spectrally selective absorbing mechanism of the layered multilayer coating, FDTD simulations were employed to investigate the interaction between solar light and the nanolaminates in the absorbing sublayer. Figure 6 depicts a cross-section of the electric field of nanoparticles with layered and random distributions at a wavelength of 527 nm, corresponding to the radiation center of the solar spectrum and indicating the widespread hot spots among the nanoparticles. Meanwhile, the electric field intensity of the nanolaminates is much higher than the structure of the nanoparticles that are randomly distributed in the sublayer. These field localizations reveal that the surface plasmon is excited at this wavelength [15,20,21].

Meanwhile, the calculated extinction spectra also reveal that the layered structure exhibits a superior light absorbing behavior than the structure wherein the nanoparticles are randomly distributed, as shown in Figure 7. In this case, the light would be held in the nanolaminates, which would strengthen the interaction intensity and time between the light and the AlCrON absorber. Thus, the as-deposited and annealing coatings show excellent solar selectivity.

In addition, the layered distribution of the AlN and Cr_2_N nanoparticles forms many Fabry–Pérot microcavities in the AlCrON absorbing sublayer. The formation of the Fabry–Pérot microcavities allows the absorption of the vast majority of the visible wavelengths (wavelength range: 390–780 nm) due to the existence of strong field intensity, as confirmed in Figure 6. For longer, near-IR wavelengths, a part of the solar light would be scattered, in addition to a part being absorbed, and it would be held in the Fabry–Pérot microcavities [22]. In this case, the interaction intensity and time between the solar light and the absorbing sublayer would be remarkably enhanced, which can boost the absorption of the solar spectra. However, the light absorption for MIR wavelengths is not enhanced, which is beneficial for lowering the emissions of the multilayer coating.

Based on the above simulations, the spectrally selective absorbing mechanism of the layered multilayer coating is summarized in Figure 8. The main absorbing mechanisms of the layered structure include localized surface plasmon resonance [19] and surface plasmon-polariton [23,24], in addition to the normal particle scattering, oscillation, and inter-band transition. When incident light enters the layered AlCrON sublayer, the nanoparticles would absorb the shortwave light due to the inter-band transition of the electrons. Meanwhile, surface plasmon polaritons and localized surface plasmon resonance would occur at the nanoparticle/amorphous interface because of the free electron oscillation and the interface effect [19,25]. In addition, there is a partial absorption that occurs in the layered AlCrON sublayer due to the scattering and the transition of the band, as well as to the vibrations of the nanoparticles [26].

### 4.2. Thermal Strengthening Mechanism

Based upon the above microstructure characterization, the formation of the nanocrystal/amorphous composite structure in the multilayer coating is considered to be the main reason for the excellent selectivity stability during long-term annealing at high temperatures in ambient air. Thus, the improved thermal stability of the AlCrON-based cermet solar selective absorbing coating was associated with an effective delay of the atomic diffusion and agglomeration.

According to our previous investigation, the annealing temperature of the amorphous matrices is lower than the crystallization temperature [27]. In this case, the amorphous matrices would remain stable during long-term annealing. In other words, the nanocrystal/amorphous structure nanolaminates can remain stable during the entire annealing treatment. As the amorphous matrices are free of grain boundaries and dislocations, the long-range diffusion of the Al and Cr atoms, as well as the inward diffusion of oxygen, can be effectively retarded. Further, the existence of the amorphous structure can alleviate the potential aggregation of nanoparticles.

Furthermore, the line-scanning and mappings shown in Figure 3 demonstrate a spatially inhomogeneous element distribution where the deficit Cr areas are rich in Al, suggesting that Al atoms would preferentially diffuse out to gather on the nanoparticles’ surfaces. These Al atoms are easily oxidized during air annealing to generate an alumina oxide layer. Thus, the AlN nanoparticles’ surfaces are partially covered by the freshly generated thin alumina layer, which can be regarded as a passivation layer and is beneficial for raising the diffusion barrier to inhibit the atomic migration so as to prevent the further oxidation of the nanoparticles. Moreover, the formation of the thin alumina layer can further suppress the coalescence of the nanoparticles via Ostwald ripening. From the complete perspective, the growth of the nanoparticles at high-temperatures cannot be ruled out, but the presence of the nanocrystal/amorphous composite structure is beneficial for isolating the absorbing particles, and so agglomeration is reduced as marginally as possible, which results in the superior thermal stability.

In conclusion, both the experimental and simulated results suggest that the introduction of lamellar-distributed nanoparticles in the cermet absorbing sublayers can enhance the thermal stability of cermet-based multilayer coatings and boost their selectivity by combining several absorbing mechanisms. This strategy is firstly proposed in the present paper, and it is identified as feasible and promising for improving the solar selectivity and thermal stability of cermet-based photothermal conversion coatings.

## 5. Conclusions

In this article, a novel stabilization strategy has been proposed to simultaneously boost the thermal tolerance and the selectivity of cermet-based solar selective absorbing coatings by forming lamellar-distributed nanoparticles in the absorbing sublayers using ion-source-assisted cathodic arc plating. The proposed coating shows a high solar absorptance of 0.924 and a low thermal emittance of 0.11, as well as an excellent spectral selectivity of 0.919/0.225 after annealing at 500 °C for 1000 h in ambient air. The unique distribution is beneficial to the absorptivity due to the formation of asymmetric Fabry–Pérot microcavities, which can enhance the interaction intensity and duration time between the absorbing sublayers and the sunlight, and thus boost the optical performance. To summarize, introducing lamellar-distributed nanoparticles into cermet sublayers can enhance solar selectivity and significantly boost thermal stability at high temperatures in ambient air, which is of great importance to the design of solar selective absorbing coatings for high-temperature photothermal conversion.

## Figures and Tables

**Figure 1 materials-15-06847-f001:**
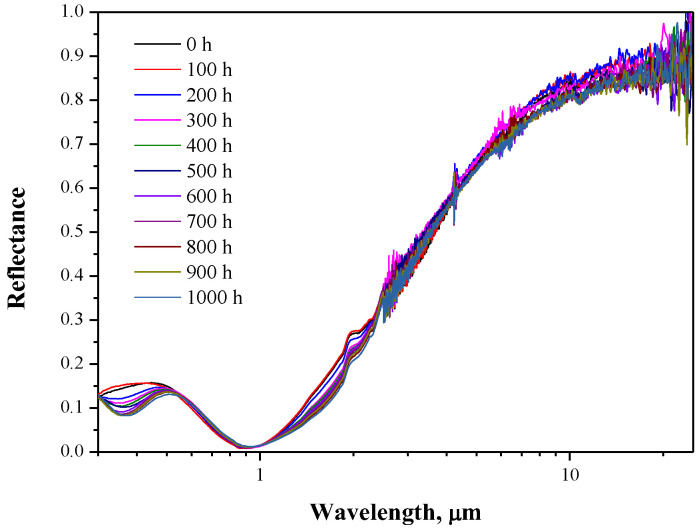
Reflectance spectra in the UV-visible-IR region of the multilayer coating before and after annealing for different times at 500 °C in ambient air.

**Figure 2 materials-15-06847-f002:**
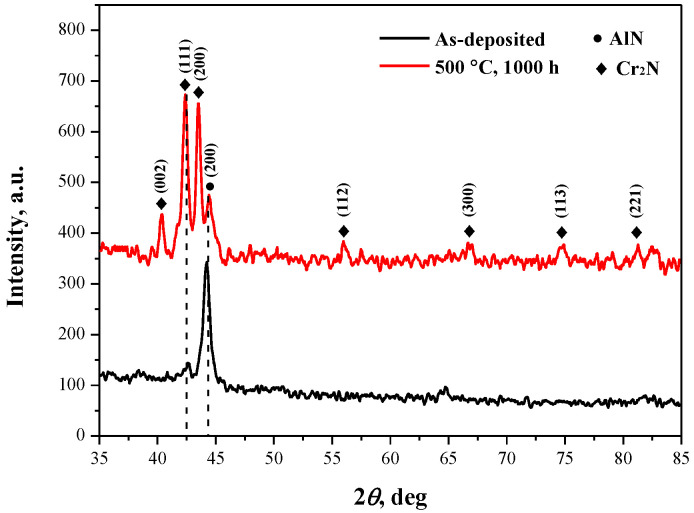
GI-XRD patterns of the coatings before and after annealing at 500 °C in ambient air for 1000 h.

**Figure 3 materials-15-06847-f003:**
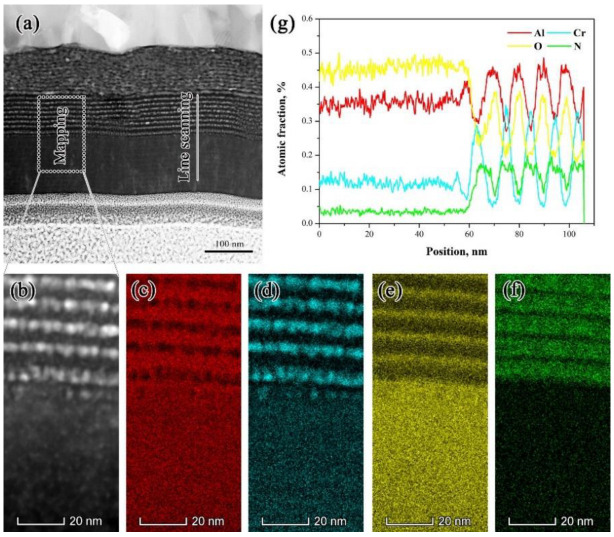
(**a**) Cross-sectional bright field TEM image of the sample annealed at 500 °C for 1000 h in ambient air. The rectangular box denotes the location of the chemical component mapping (**b**–**f**), and the EDS line scan data are shown in panel (**g**).

**Figure 4 materials-15-06847-f004:**
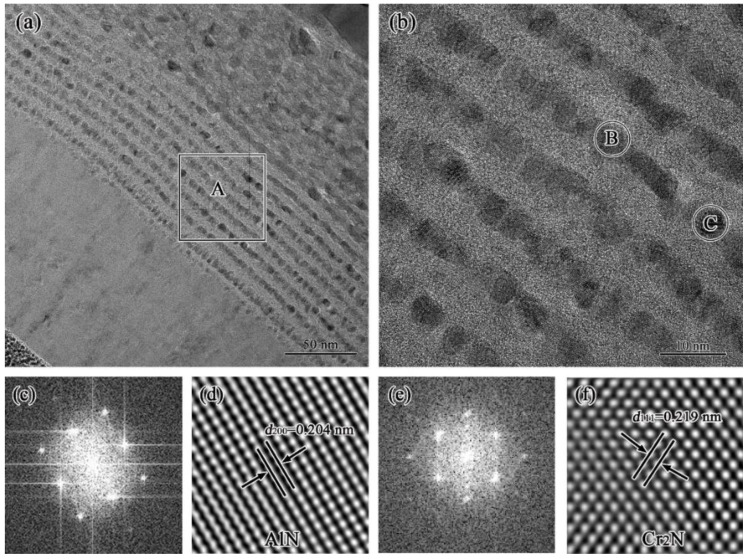
(**a**) HAADF-STEM image of the annealed sample in which the rectangular box denotes the location of the HR-TEM analysis shown in (**b**), (**c**,**e**) are the FFT (fast Fourier transform) images of the areas B and C as denoted in (**b**), respectively, and (**d**,**f**) are the IFFT (inverse fast Fourier transform) images of the areas B and C as denoted in (**b**), respectively.

**Figure 5 materials-15-06847-f005:**
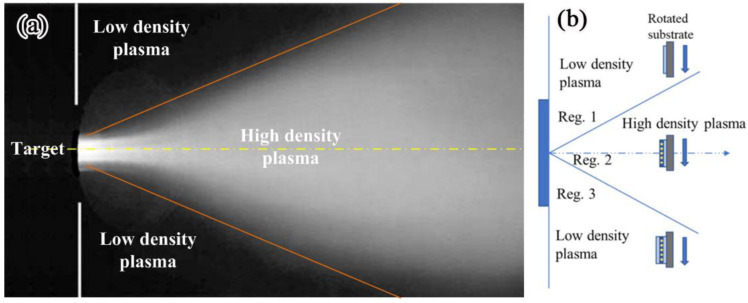
Plasma pluming from the flange-mounted source into the chamber (**a**), and a schematic diagram of the deposition of the layered nanoparticle coating (**b**).

**Figure 6 materials-15-06847-f006:**
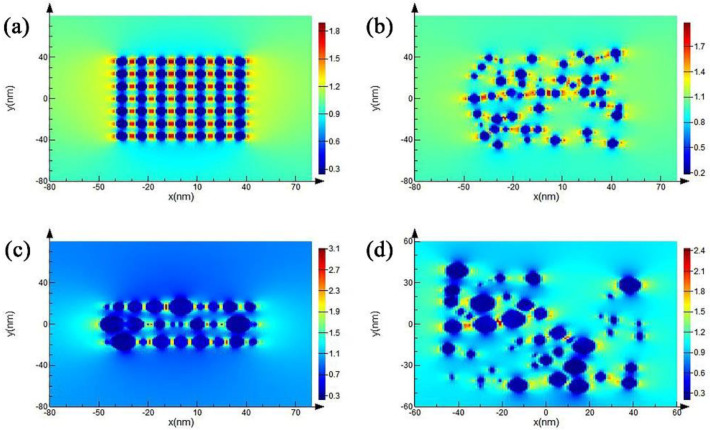
Cross-section of the electric field distributions of the layered (**a**) and randomly distributed (**b**) nanoparticles of the same diameter, and the layered (**c**) and randomly distributed (**d**) nanoparticles with different diameters, at a wavelength of 527 nm.

**Figure 7 materials-15-06847-f007:**
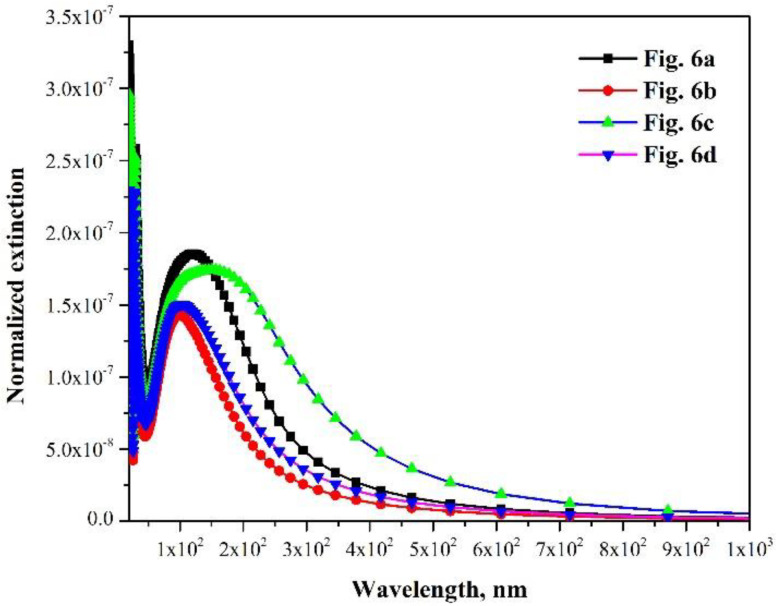
The calculated extinction spectra of the models shown in Figure 6.

**Figure 8 materials-15-06847-f008:**
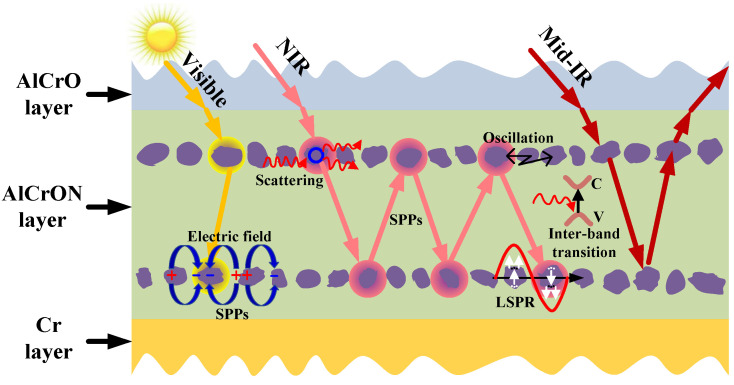
Schematic map of the selective absorbing mechanism behind the multilayer cermet coating with nanolaminates in the absorbing sublayer.

**Table 1 materials-15-06847-t001:** The calculated absorptance and emittance of the as-deposited and annealed multilayer coatings.

Annealing Time, h	α	ε	α/ε	PC
0	0.909	0.163	5.58	-
100	0.911	0.150	6.07	−0.0085
200	0.913	0.144	6.34	−0.7785
300	0.914	0.157	5.82	−0.008
400	0.916	0.180	5.09	0.0015
500	0.917	0.180	5.09	0.0005
600	0.918	0.180	5.10	−0.0005
700	0.919	0.183	5.02	0
800	0.920	0.185	4.97	0
900	0.923	0.185	4.99	−0.003
1000	0.922	0.185	4.98	−0.002

## Data Availability

The data presented in this study are available on request from the corresponding author.

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
