# Peer review of "A Feasible and Promising Strategy for Improving the Solar Selectivity and Thermal Stability of Cermet-Based Photothermal Conversion Coatings"

_materials, 2022, doi:10.3390/ma15196847_

Round 1

Reviewer 1 Report

In this work the authors discuss a novel cermet AlCrON/AlCrN/multilayer terminated with AlCrO antireflective coating and with a base of Cr IR reflector, useful for solar thermo-photovoltaics application.

AlN and Cr2N nanocrystals layers are intercalated with amorphous AlCrON layer. The Zener pinning effect at the crystal-amorphous layer interface hinders the coalescence of nitride nanoparticles and so this kind of structure yields high temperature resistance even after air environment annealing at high temperatures (500°C) and after multiple prolonged annealing/cooling cycles along with a high optical absorbance in the UV-Vis and IR wavelength range (solar spectrum range) and low thermal emittance. 

The 'nanolaminates' structure also contribute to electric field enhancement and trapping inside the structure itself boosting the solar light harvesting capability by varoius described mechanisms: interband transitions, LSPR, SPP, IR scattering and absorption, conversion into vibrational/oscillation modes of nanoparticles...

- the exposition of the procedures, techniques and the discussion of the advantages of these 'nanolaminates' are quite well exposed, there are only very minor quirks in the text editing that I have noted in the updated revised pdf 

- the caption of figure 4 must improved a bit in terms of clarity for the reader

- I have a perplexity on the conclusions, because throughout the article the annealing temperature is kept at 500°C, while in the conclusions results after 2000h (10 cycles) at 600°C are given for solar absorptance and thermal emittance, not mentioned in the body of the paper before. 

Kind Regards

Author Response

Detailed response to Reviewer #1:

In this work the authors discuss a novel cermet AlCrON/AlCrN/multilayer terminated with AlCrO antireflective coating and with a base of Cr IR reflector, useful for solar thermo-photovoltaics application.

AlN and Cr2N nanocrystals layers are intercalated with amorphous AlCrON layer. The Zener pinning effect at the crystal-amorphous layer interface hinders the coalescence of nitride nanoparticles and so this kind of structure yields high temperature resistance even after air environment annealing at high temperatures (500°C) and after multiple prolonged annealing/cooling cycles along with a high optical absorbance in the UV-Vis and IR wavelength range (solar spectrum range) and low thermal emittance. 

The 'nanolaminates' structure also contribute to electric field enhancement and trapping inside the structure itself boosting the solar light harvesting capability by varoius described mechanisms: interband transitions, LSPR, SPP, IR scattering and absorption, conversion into vibrational/oscillation modes of nanoparticles.

- the exposition of the procedures, techniques and the discussion of the advantages of these 'nanolaminates' are quite well exposed, there are only very minor quirks in the text editing that I have noted in the updated revised pdf.

Resoponse: Special thanks to you for your careful and nice revisions. We have corrected the manuscript accordingly.

Revisions made:

The samples were cooled down to room temperature naturally for the optical measurements after an interval of 100 h (denoted as one cycle in the present study), then followed by the repeating annealing cycle.

The cross-sectional TEM images of the multilayer coating were acquired using transmission electron microscopy (TEM, Tecnai G2 F20 S-TWIN, USA) to identify the microstructural features of the multilayer coating.

In addition, the broadband fixed angle source technique was employed in FDTD simulations.

It was worth noting that no oxide nanocrystals were detected in the AlCrON absorbing sublayer although we carried out abundant HR-TEM tests. The absence of oxide nanocrystals may be related to its strong tendency of forming amorphous structure, which merits further study.

The plasma density distribution is more peaked in the axial direction as shown in Fig. 5a due to the applied of axial magnetic field.

In this case, the aluminum ions can completely react with the reactive gas, which would result into the formation of amorphous matrix in each sublayer.

The coating shows a high solar absorptance of 0.924 and a pretty low thermal emittance of 0.11, as well as an excellent spectral selectivity of 0.919/0.225 after at 500 °C for 1000 h in air.

- the caption of figure 4 must improved a bit in terms of clarity for the reader

Resoponse: Thanks for your kind suggestion, we have corrected the the caption of figure 4.

Revisions made:

Figure 4. (a) HAADF-STEM image of the annealed sample, in which the rectangular box denotes the location of HR-TEM analysis shown in 4b, (c) and (e) are the FFT (Fast Fourier Transform) images of the areas B and C as denoted in 4b, respectively, while (d) and (f) are the IFFT (Inverse Fast Fourier Transform) images of the areas B and C as denoted in 4b, respectively.

- I have a perplexity on the conclusions, because throughout the article the annealing temperature is kept at 500°C, while in the conclusions results after 2000 h (10 cycles) at 600°C are given for solar absorptance and thermal emittance, not mentioned in the body of the paper before. 

Resoponse: I am really sorry for this mistake. The reason for this mistake is because we had done anealing treatment experiments at 600°C for 2000 h (10 cycles, the interval is 200 h). According to your kind suggestion, we have corrected this mistake in the revised manuscript.

Revisions made:

The coating shows a high solar absorptance of 0.924 and a pretty low thermal emittance of 0.11, as well as an excellent spectral selectivity of 0.919/0.225 after at 500 °C for 1000 h in air.

Reviewer 2 Report

Thanks are due to the authors for presenting their good work.

Please address the following comments.

The authors have introduced their work by stating “In this work, we proposed a novel strategy 72 that can simultaneously stabilize the nanoparticles and boost the selectivity by introducing lamellar distributed nanoparticles in the absorbing sublayers using ion source assisted 74 cathodic arc plating.” The novel strategy has been introduced in the authors’ previous work [9]", Please update the statement to reflect the advance in work at this stage compared to the earlier stage.

Please also clarify if this work is based on developing new samples, or a further study on the old samples.

I have noted that in this work annealing is reported to occur at 500°C, while in previous work it was at 550°C.

Please explain the suggested reasons for the formation of the multilayer structure consisting of Cr IR reflector, AlCrN/AlCrON absorbers and AlCrO anti-reflective sublayer from the substrate to the surface, as shown in Fig. 3a. According to the classical theory, lamellar structures result from control of cooling, and growth conditions, so how do you explain the lamellar structures forming after annealing? Please, also confirm the appearance of lamellar structures in the as- deposited condition as Fig. 3 refers to the annealed condition. This would have an effect on the statement “The formation of nanolaminates in the multilayer coating is highly related to the 211 characteristic of cathodic arc plating” if the lamellar structures only appeared after annealing.

In several locations, the word amorphous appears on its own. Please update to be amorphous structures, wherever it appears.

In the conclusions, the authors mention” after at 600 °C for 2000 h in air”, whereas this condition is not presented in this work.

Author Response

Detailed response to Reviewer #2:

1. The authors have introduced their work by stating “In this work, we proposed a novel strategy that can simultaneously stabilize the nanoparticles and boost the selectivity by introducing lamellar distributed nanoparticles in the absorbing sublayers using ion source assisted cathodic arc plating.” The novel strategy has been introduced in the authors’ previous work [9]", Please update the statement to reflect the advance in work at this stage compared to the earlier stage.

Resoponse: Indeed, this kind of microstructure had been reported in our previous work [9]. However, the idea that lamellar distributed nanoparticles can simultaneously stabilize the nanoparticles and boost the selectivity was firstly proposed in the present work. Meanwhile, the absorbing behavior of nanolaminates was similuated, which can combine several spectrally selective absorbing mechanisms. Considering your question, we have discussed this question in the revised manuscript.

Revisions made:

In conclusion, both the experimental and simulated results suggest that the introduction of lamellar distributed nanoparticles in the cermet absorbing sublayers not only can enhance the thermal stability of cermet-based multilayer coating, but also boost the selectivity by combining several absorbing mechanisms. This strategy is firstly proposed in the present paper and is identified feasible and promising for improving solar selectivity and thermal stability of cermet-based photothermal conversion coatings.

2. Please also clarify if this work is based on developing new samples, or a further study on the old samples.

Resoponse: This work is a further study on the old samples. The samples used in the present study were the same batch of our previous study. In Section 2.1, We have clarified the deposition parameters are described in the previous work [13].

3. I have noted that in this work annealing is reported to occur at 500°C, while in previous work it was at 550°C.

Resoponse: Frankly speaking, we have done the thermal stability tests at different annealing temperature, such as 500°C, 550°C, 600°C and 650°C. However, the emphasis of different temperatures is different. For example, the main aim of annealing at 500°C is to test the thermal stability of nanolaminates while 550 °C is to evaluate the thermal stability of nanoparticles. Considering your question, we have added the reason for annealing temperature selection.

Revisions made:

For the annealing procedure, 5 pieces of coating were set in the center of the muffle furnace and then were heated to 500 °C with a heating rate of 10 °C/min. The temperature accuracy of the muffle furnace was ± 5 °C. The thermal stability of the multilayer coating is mainly determined by the coarsening behavior of nanoparticles and the stability of structure in the abosrbing sublayer. In the present study, the aim is to test the thermal stability of AlCrON and AlCrN sublayer.

4. Please explain the suggested reasons for the formation of the multilayer structure consisting of Cr IR reflector, AlCrN/AlCrON absorbers and AlCrO anti-reflective sublayer from the substrate to the surface, as shown in Fig. 3a. According to the classical theory, lamellar structures result from control of cooling, and growth conditions, so how do you explain the lamellar structures forming after annealing? Please, also confirm the appearance of lamellar structures in the as-deposited condition as Fig. 3 refers to the annealed condition. This would have an effect on the statement “The formation of nanolaminates in the multilayer coating is highly related to the characteristic of cathodic arc plating” if the lamellar structures only appeared after annealing.

Resoponse: Firstly, the lamellar structures were formed during the as-deposited condition rather than the annealing process. As can be seen from Fig. 1, the reflectance curves before and after annealing are basically the same. Thus, the multilayer coating cannot experience big microstructure changes during annealing. Also, our previous study on the as-deposited multilayer suggest that the microstructure is similar with the annealed sample. The formation mechanism of nanolaminates in the multilayer coating is discussed in the last paragraph of Page 7.

5. In several locations, the word amorphous appears on its own. Please update to be amorphous structures, wherever it appears.

Resoponse: Corrected accordingly.

Revisions made:

Also, the existence of amorphous structure can alleviate the potential aggregation of nanoparticles.

It is worth noting that the nanoparticles are randomly distributed in the amorphous structured cermet matrices, which facilitates the agglomeration at high-temperatures, and thus accelerates the deterioration of cermet sublayers.

The absence of oxide phases was attributed to their amorphous structure nature as suggested in our previous paper [9].

In other words, the nanocrystal/amorphous structure nanolaminates can keep stable during the whole annealing treatment.

6. In the conclusions, the authors mention” after at 600 °C for 2000 h in air”, whereas this condition is not presented in this work.

Resoponse: Actually, we have corrected this statement in the revised version. You can download the latest manuscript.

Reviewer 3 Report

This paper reports on a novel multilayer solar selective absorbing coating containing lamellar distributed nanoparticles in the cermet absorbing sublayers and fabricated by ion source assisted cathodic arc plating. The coating demonstrates enhanced selectivity and thermal stability attributed to the formation of lamellar distributed nanoparticles in the absorbing sublayer, which forms many asymmetric Fabry-Pérot cavities. The optical properties and microstructure of the multilayer coating has been investigated. Much attention is given to the analysis of the spectrally selective absorbing mechanism of the layered multilayer coating as well as the thermal strengthening mechanism.

The manuscript suits the topics of Materials and can be interesting for the readership of the journal as it presents a promising strategy for improving solar selectivity and thermal stability of cermet-based photothermal conversion coatings. I would recommend the acceptance of the manuscript after the following revision:

1. Line 91. You write “The samples were cooled down to room temperature naturally for the optical measurements after an interval of 200 h (one cycle in our case), then followed by the repeating annealing cycle.” It seems one cycle was equal to 100 h.

2. Line 140. You write “The as-deposited multilayer coating achieves near-perfect sunlight reflection over the wavelength of 0.3-1.5 μm.” Why do you declare that reflection is near-perfect, if it is less than 15 percent?

3. In Table 1. Holding time should be changed to Annealing time.

4. Line 267. The statement “For visible wavelengths, the light absorption is mainly absorbed” is not clear.

5. The conclusions are not based on the results of the research. During the tests, the fabricated coating was annealed for up to 1000 h at a temperature of 500 °C, and the conclusions refer to the results obtained after annealing the coating for 2000 h at a temperature of 600 °C.

Author Response

Detailed response to Reviewer #3:

This paper reports on a novel multilayer solar selective absorbing coating containing lamellar distributed nanoparticles in the cermet absorbing sublayers and fabricated by ion source assisted cathodic arc plating. The coating demonstrates enhanced selectivity and thermal stability attributed to the formation of lamellar distributed nanoparticles in the absorbing sublayer, which forms many asymmetric Fabry-Pérot cavities. The optical properties and microstructure of the multilayer coating has been investigated. Much attention is given to the analysis of the spectrally selective absorbing mechanism of the layered multilayer coating as well as the thermal strengthening mechanism.

The manuscript suits the topics of Materials and can be interesting for the readership of the journal as it presents a promising strategy for improving solar selectivity and thermal stability of cermet-based photothermal conversion coatings. I would recommend the acceptance of the manuscript after the following revision:

  1. Line 91. You write “The samples were cooled down to room temperature naturally for the optical measurements after an interval of 200 h (one cycle in our case), then followed by the repeating annealing cycle.” It seems one cycle was equal to 100 h.

Resoponse: I am sorry for this mistake. Indeed, one cycle was equal to 100 h as shown in Fig. 1 and Table 1 that the data sequence arrives to 1000 hours. According to your kind suggestion, we have corrected this mistake in the revised manuscript.

Revisions made:

The samples were cooled down to room temperature naturally for the optical measurements after an interval of 100 h (denoted as one cycle in the present study), then followed by the repeating annealing cycle.

  1. Line 140. You write “The as-deposited multilayer coating achieves near-perfect sunlight reflection over the wavelength of 0.3-1.5 μm.” Why do you declare that reflection is near-perfect, if it is less than 15 percent?

Resoponse: I am sorry for this confusing statement. For perfect sunlight absorbing surface, the reflectance in 0.3-2.5 μm should be zero. However, perfect solar selective absorbing coating does not exist. The reflectance curves of the multilayer coatings in the present study are as low as 0.05 near wavelength of 0.9 μm. Considering your question, we have corrected this sentence accordingly.

Revisions made:

The as-deposited multilayer coating shows relatively low sunlight reflection over the wavelength of 0.3-1.5 µm, which corresponding to the central zone of solar radiation.

  1. In Table 1. Holding time should be changed to Annealing time.

Resoponse: Corrected accordingly.

Revisions made:

Table 1

  1. Line 267. The statement “For visible wavelengths, the light absorption is mainly absorbed” is not clear.

Resoponse: We have rewritten this sentence.

Revisions made:

The foramtion of Fabry-Pérot microcavities can absorb the vast majority of visible wavelengths (wavelength range: 390 - 780 nm) due to the existence of strong field intensity as confirmed in Fig. 6.

  1. The conclusions are not based on the results of the research. During the tests, the fabricated coating was annealed for up to 1000 h at a temperature of 500 °C, and the conclusions refer to the results obtained after annealing the coating for 2000 h at a temperature of 600 °C.

Resoponse: I am really sorry for this mistake. The reason for this mistake is because we had done anealing treatment experiments at 600°C for 2000 h (10 cycles, the interval is 200 h). According to your kind suggestion, we have corrected this mistake in the revised manuscript.

Revisions made:

The coating shows a high solar absorptance of 0.924 and a pretty low thermal emittance of 0.11, as well as an excellent spectral selectivity of 0.919/0.225 after at 500 °C for 1000 h in air.